# Emerging Roles of Ephexins in Physiology and Disease

**DOI:** 10.3390/cells8020087

**Published:** 2019-01-24

**Authors:** Kwanhyeong Kim, Sang-Ah Lee, Daeho Park

**Affiliations:** 1School of Life Sciences and Aging Research Institute, Gwangju Institute of Science and Technology, Gwangju 61005, Korea; sanga03@gist.ac.kr; 2Research Center for Cellular Homeostasis, Ewha Womans University, Seoul 03760, Korea

**Keywords:** Ephexin, Ephexin1, Ephexin2, Ephexin3, Ephexin4, Ephexin5, guanine nucleotide exchange factor, GEF, Dbl family, Rho GTPase

## Abstract

Dbl (B-cell lymphoma)-related guanine nucleotide exchange factors (GEFs), the largest family of GEFs, are directly responsible for the activation of Rho family GTPases and essential for a number of cellular events such as proliferation, differentiation and movement. The members of the Ephexin (Eph-interacting exchange protein) family, a subgroup of Dbl GEFs, initially were named for their interaction with Eph receptors and sequence homology with Ephexin1. Although the first Ephexin was identified about two decades ago, their functions in physiological and pathological contexts and regulatory mechanisms remained elusive until recently. Ephexins are now considered as GEFs that can activate Rho GTPases such as RhoA, Rac, Cdc42, and RhoG. Moreover, Ephexins have been shown to have pivotal roles in neural development, tumorigenesis, and efferocytosis. In this review, we discuss the known and proposed functions of Ephexins in physiological and pathological contexts, as well as their regulatory mechanisms.

## 1. Introduction

The Rho family of GTPases is a subgroup of the Ras superfamily. Like other small GTPases, the members of the Rho family function as molecular switches [1,2,3,4]. Rho GTPases are best known for their roles in the regulation of the actin cytoskeleton. Consequently, they are involved in diverse cytoskeleton-dependent processes such as cell adhesion, cell motility, cytokinesis, phagocytosis, morphogenesis, and axon guidance [1,2,5]. In addition, Rho GTPases also play essential roles in transcriptional regulation and cell transformation [1,2,5,6,7]. Because Rho GTPases are associated with these essential cellular processes, the activity of Rho GTPases should be spatiotemporally and tightly controlled. Accordingly, various regulatory molecules are involved to modulate the activity of Rho GTPases, but guanine nucleotide exchange factors (GEFs), GTPase-activating proteins (GAPs), and guanine nucleotide dissociation inhibitors (GDIs) are at the center of the regulation.

The overall activity of a Rho GTPase in cells depends on its ratio of the GTP-bound to GDP-bound form. Importantly, this ratio is controlled by its direct binding to GEFs, GAPs, or GDIs. GEFs activate Rho GTPases by catalyzing the exchange of bound GDP for GTP, which results in activated Rho GTPases to interact with their downstream effectors. On the other hand, GAPs antagonize the function of GEFs by stimulating Rho GTPases to hydrolyze bound GTP to GDP. Additionally, inactive GDP-bound Rho GTPases are sequestered in the cytosol by RhoGDIs (Figure 1) [1,2,8,9,10].

Dbl, the first GEF of the Dbl family, was originally identified as an oncogene but its function as a GEF for Cdc42 was revealed later [11,12]. Since then, 82 RhoGEFs have been identified; 71 RhoGEFs comprise the Dbl family and the other 11 RhoGEFs belong to the Dock family [13]. An important characteristic of the Dbl family is the presence of the Dbl homology (DH) domain, which is composed of ~200 amino acid residues, and the pleckstrin homology (PH) domain, which is composed of ~100 residues [9,10,14]. The DH domain is mainly responsible for the catalytic activity of Dbl RhoGEFs and forms a minimal catalytic unit for the guanine nucleotide exchange reaction with the adjacent PH domain [9,14,15]. In addition to the tandem DH-PH domain, the members of the family also contain other domains that regulate their GEF activity, subcellular localization, or interaction with other molecules [9]. In contrast, the GEFs of the Dock family are considered as unconventional GEFs due to the absence of the typical DH domain. Thus, the Dock family, instead of the DH domain, is characterized by the presence of DHR1 (Dock homology region 1) and DHR2 (Dock homology region 2) whose roles are to bind phospholipids and to provide the guanine nucleotide exchange activity, respectively [15,16]. Intriguingly, the number of RhoGEFs far outnumbers that of Rho GTPases. Moreover, some RhoGEFs can modulate multiple Rho GTPases. These imply that the activity of Rho GTPases is modulated by intertwined networks [14,15].

Among the members of the Dbl family, five proteins from Ephexin1 to 5 belong to the Ephexin subfamily [13,17]. Their sequences are highly conserved among paralogs, and they possess the typical tandem DH-PH domain and an additional SH3 domain which is C-terminal to the PH domain except that the SH3 domain is absent in Ephexin5 [17,18]. Besides the high sequence homology, one of the important common features of the Ephexin family is that they are the direct downstream proteins of Eph receptors, the largest subfamily of receptor tyrosine kinases that is activated by Ephrins and involved in various cellular processes such as axon guidance, formation of tissue boundaries, long-term potentiation, angiogenesis, and cancer, through their association with Eph receptors [18,19,20,21,22,23]. Therefore, the GEF activity of Ephexins could be regulated by Ephrin/Eph receptor-mediated signaling, and diverse cellular processes induced by Ephexins occur through this module, Ephrin-Eph receptor-Ephexin (Figure 2).

In particular, the Ephexin family plays essential roles in normal function of neurons and their development. It regulates the axon guidance of developing neurons [17,19,24,25,26,27,28,29,30] and synaptic homeostasis [31,32,33] via the reorganization of the actin cytoskeletons. Although a number of studies on the family have focused on its roles in the nervous system, several studies have also suggested that Ephexins have a variety of other functions. For example, it has been reported that they regulate angiogenesis [34], efferocytosis, a type of phagocytosis to clear apoptotic cells in the body [35,36,37], and vascular muscle contractility [18]. Moreover, aberrant expression or function of Ephexins are related to various types of cancer [38,39,40,41,42,43,44,45,46,47,48,49,50,51,52,53]. Thus, the roles of Ephexins are not limited to the nervous system. In this review, we highlight the molecular characteristics of Ephexins and their various roles in physiological and pathological contexts.

## 2. Ephexin Family

Since the first Ephexin, Ephexin1, was identified as an EphA4-interacting RhoGEF, Ephexin was named for Eph-interacting exchange protein [19]. Since then, four Ephexin1-homologous proteins have been identified and named sequentially from Ephexin2 to Ephexin5 [17]. These five Ephexins comprise a subfamily of the Dbl-related GEFs, the Ephexin family [13,17].

Ephexins function as GEFs for Rac, Cdc42, Rho and RhoG, and interact with Eph receptors to transduce signals from the receptors although an Eph receptor interacting with Ephexin2 has not been identified. Accordingly, they are engaged in a variety of Eph receptor-mediated cellular processes (Figure 2 and Table 1) [18,19,20,21,22]. In addition, the domain structure of Ephexins is highly conserved, which consists of a tandem DH-PH domain followed by a SH3 domain, but the length of their N-termini is irregular and the sequence identity is especially low among Ephexin1 orthologs, particularly drosophila and chicken Ephexin1 (dEpehxin1 and cEphexin1) (Figure 3). In the following sections, the members of the Ephexin family will be consecutively discussed in detail (also refer to the Appendix A to see all experimental systems for studies on Ephexins).

### 2.1. Ephexin1

Ephexin1 was originally isolated from an adult mouse brain cDNA library in 2000 as Ngef. Ephexin1 is predominately expressed in the central nervous system, the brain and spinal cord, and its expression is developmentally regulated, that is, its mRNA levels gradually increase throughout embryonic development and peak at the P10 postnatal stage. In particular, Ephexin1 is highly expressed in the caudate nucleus associated with motor processes [19,54]. This expression timing and location of Ephexin1 is quite similar to those of EphA4, which suggests the implicated roles of Ephexin1 in the nervous system. Indeed, Ephexin1 plays crucial roles in axon guidance and synaptic homeostasis.

During development, the axon pathfinding is regulated by various molecules existing in the extracellular matrix or on the surrounding cell surfaces. These guidance cues determine the attraction and/or repulsion of the growth cone by regulating the actin cytoskeleton. Eph receptors and Ephrins are involved and their roles are well established in these processes [58]. In particular, Ephexin1 is a key regulator providing a linkage between Ephrin-Eph and the axon pathfinding through interaction of Ephexin1 with EphA4 [19]. Furthermore, Ephexin1 participates in mediating proper neuronal functions like synaptic homeostasis, maturation and myelination [33,59]. In the neuromuscular junction (NMJ) of fruit flies, presynaptic Eph receptors receive a retrograde signal from the postsynaptic terminal and activate dEphexin1, Cdc42, and calcium channels sequentially, leading to the enhanced presynaptic release for synaptic homeostasis [31]. In mouse NMJ, Ephexin1 expressed in the postsynaptic muscle regulates the actin cytoskeleton via RhoA. As a consequence, the membrane structure and postsynaptic AchR cluster stability are altered and thus NMJ maturation is mediated [32].

In addition, the importance of Ephexin1 in the nervous systems is also highlighted in various contexts. Optimal neuron innervation by axon guidance during embryonic development is one of the most well-characterized processes regulated by Ephexin1. Tissues from various model organisms like mice, chickens, and fruit flies have been used to confirm the roles of Ephexin1 in those developmental stages. Afferent innervation from spiral ganglion neurons (SGNs) is important for development of the proper auditory system during mouse embryogenesis and also regulated by the ephrin-EphA4-Ephexin1 axis [26]. Medial lateral motor column (LMC) neurons and dorsal limb mesenchyme express EphB1 and Ephrin-B, respectively, to induce the growth of medial LMC neurons towards the ventral part in chicken and mouse embryos [27]. dEphexin1 also mediates olfactory dendrite targeting [29] and cEphexin1 (chicken Ephexin1) takes an important role in successful retinal ganglion cell (RGC) projection to reach the optic tectum in chicken embryos [28].

The GEF activity of Ephexin1 is modulated by the activation state of EphA4: using cultured Ephexin1-/- neurons and RNA interference in the chick, it was found that Ephexin1 could function as a GEF for Rac1, Cdc42, and RhoA under the basal condition. However, when EphA4 is activated by Ephrins, the phosphorylation of Ephexin1 by Src provides the specificity of Ephexin1 towards RhoA [17,24]. Ahead of this modification, Cdk5 participates in the phosphorylation of Ephexin1 as a priming kinase. Activated EphA4 phosphorylates Cdk5, which causes Cdk5 to phosphorylate Ephexin1, results in further modification of Ephexin1 by Src [25]. Intriguingly, this phosphorylation of Ephexin1 relieves auto-inhibition generated by the inhibitory helix region to the N-terminus of the DH domain of Ephexin1. It is known that the activity of a number of Dbl family GEFs is auto-inhibited by interaction of the DH domain with an N-terminal helix region to exclude and prevent the activation of Rho GTPases. Similarly, Ephexin1 is also auto-inhibited and its activity is modulated by relief or strength of the auto-inhibition. The phosphorylation of tyrosine 179, located in the inhibitory helix region of Ephexin1, by Src disrupts interaction between the DH domain and the inhibitory helix region of Ephexin1, which makes the DH domain free to bind RhoA. Eventually, the phosphorylation increases the activity of Ephexin1 and the levels of RhoA activation [60].

It has been reported that Ephexin1 is positively correlated to pathophysiologic conditions like depression or recovery after CNS (central nervous system) injury, due to its roles in neuronal development and synaptic homeostasis, [61,62,63,64].

### 2.2. Ephexin2

Ephexin2 was identified in 2004 from a mouse liver cDNA library and initially was termed as Wgef, an abbreviation of weakly similar to Rho GEF 5 [55]. Studies on Ephexin2 have not extensively been done compared with Ephexin1. Thus, there is not much literature to refer to and data about Ephexin2 are also limited. However, its intrinsic role to function as a GEF for RhoA seems to be clear.

Ephexin2 is involved in convergent extension, a developmental step of anterior-posterior axis extension in Xenopus gastrulation through RhoA activation [65]. In addition, Ephexin2 participates in pronephric tubulogenesis of Xenopus and zebrafish. During this process, the GEF activity of Ephexin2 is modulated by Daam1 interacting with it [66]. Ephexin2 also plays a role in cancer. For instance, various cancer tissues and cell lines show elevated levels of Ephexin2, which results in the increased activity of RhoA causing higher cancer proliferation, migration, and invasion. The downregulated levels of Ephexin2 by miR-503 have an inhibitory effect on cancer metastasis in hepatocellular carcinoma model and miR-29b, which downregulates the levels of Ephexin2, is decreased in non-small cell lung cancer (NSCLC) [39,40]. Interestingly, the roles of Ephexin2 during pronephric tubulogenesis and in the cancer are relevant to the high expression levels of Ephexin2 in the kidney, as well as the liver and lung.

### 2.3. Ephexin3

Ephexin3, also called Tim (Transforming immortalized mammary), is ubiquitously expressed in many tissues, such as colon, kidney, trachea, prostate, liver, and pancreas, with tendency to be highly expressed in tissues containing epithelial cells [41]. Ephexin3 is able to activate RhoA in in vitro guanine nucleotide exchange assays using purified proteins and in Rho GTPase pull-down assays using COS-7 cells [20,41]. However, it promotes the formation of membrane ruffles and filopodia and induces a loss of actin stress fibers in NIH/3T3 and COS-7 cells, which seems to indicate that Ephexin3 activates Rac and Cdc42 [44]. However, the consecutive studies on Ephexin3 have shown that it functions as a GEF for RhoA rather than Rac and Cdc42.

The roles of Ephexin3 in physiological and pathological contexts are quite various. Genetic deletion of Ephexin3 causes defects in immature dendritic cell migration in vivo. In terms of cell migration, Ephexin3 is also involved in Src-induced podosome formation which is related to cell migration and adhesion. RNAi-mediated knockdown of Ephexin3 inhibits Src-dependent podosome formation whereas its overexpression increases the podosome formation through RhoA activation [45,67]. In addition, Ephexin3 is highly linked to cancer progression. Higher levels of Ephexin3 are observed in cervical, colorectal, and lung-derived cancers and patients with a high level of Ephexin3 and Src show shorter survival time [41,42,43,45,46]. Moreover, mutations in Ephexin3 is also correlated with cancer. Frequent mutations in Ephexin3 in lung cancer and familial lung cancer compared with the healthy control were detected through whole genome sequencing [68]. Furthermore, it is conceivable that the correlation of Ephexin3/Src with cancer malignancy comes from their function of promoting endothelial-mesenchymal transition (EndoMT) [47].

The activity of Ephexin3 is also auto-inhibited by intramolecular interaction caused by the N-terminal inhibitory helix and the DH domain, which is similar to that of Ephexin1. Therefore, the regulatory mechanism for the activity of Ephexin3 is also comparable to that of Ephexin1. The inhibitory helix of Ephexin3 binds to the DH domain, which prevents RhoA access to the DH domain of Ephexin3 and results in the auto-inhibition. Tyrosine phosphorylation at Y1097 and Y1100 by Src leads to disruption of the intramolecular interaction between the inhibitory helix and the DH domain, which results in RhoA activation [20,69,70].

### 2.4. Ephexin4

Most studies on Ephexin4, also known as Arhgef16, have focused on its role as a GEF for RhoG. However, the potential activity of Ephexin4 to regulate Cdc42 was also reported in a specific context. Tax-interacting-protein 1 (Tip-1) is a protein that interacts with HPV16 E6 oncoprotein to regulate E6-dependent cell motility. Intriguingly, Ephexin4 was identified as a novel binding partner of Tip-1 via a yeast two-hybrid screen. The interaction of Ephexin4 with Tip-1 alters its GEF activity toward Cdc42 [56,57].

The biological significance of Ephexin4 in various physiological and pathological conditions has been addressed. The potential involvement of Ephexin4 in preventing carcinogenesis was suggested from its down-regulation in kojic acid-stimulated A375 malignant melanoma cells [48]. In addition, it was reported that Ephexin4 could modulate cancer cell migration in a breast cancer model. Ephexin4 activates RhoG by interacting with EphA2, which promotes RhoG/ELMO2/Dock4 complex formation resulting in Rac activation [21]. Since elevated EphA2 expression is correlated with the aggressiveness of breast cancer, Ephexin4 links EphA2 to RhoG causing cancer invasion. Cancer cells are resistant to anoikis, a type of cell death caused by detachment from surrounding ECM, leading to more progression of cancer. It has been revealed that Ephexin4 takes an important part in preventing anoikis as its loss in HeLa cells promoted anoikis. Phosphorylation of EphA2 enhances its interaction with Ephexin4 and, in turn, activates RhoG and RhoG-dependent PI3K/Akt signaling. Because Akt phosphorylates EphA2, these proteins compose a positive feedback loop for anoikis resistance [50,51,52]. Ephexin4 also correlates with the development and progression of brain tumors [49,53]. In patients with oligodendroglial brain tumors, a missense mutation (2125G to 2125A) in Ephexin4 is commonly found, although extensive investigation is required to fully elucidate its contribution to the pathogenesis [49]. One recent report revealed that GLI2, glioma-associated oncogene and a downstream effector of Hedgehog signaling, binds to the promoter region of Ephexin4 and upregulates the transcript level of Ephexin4. Although there is an absence of studies on its mechanism, interaction between Ephexin4 and cytoskeleton-associated protein 5 (CKAP5) in this model is newly reported. This suggests a correlation between spindle regulation and glioma proliferation and migration [53]. Besides tumorigenesis, Ephexin4 is related to optic nerve regeneration process in zebrafish. After optic nerve injury, Wnt-related pathways are altered and the transcription levels of Daam1 and Ephexin4 decrease as a consequence [71].

A novel role of Ephexin4 in clearance of apoptotic cells was recently reported [35]. Interaction of Ephexin4 and Elmo1 was newly identified through a yeast two-hybrid screen. Elmo1 is known to regulate various cellular processes, such as cell motility, neurite outgrowth, and formation of cellular protrusions, all of which are mediated by the actin cytoskeleton remodeling. Elmo1 plays a central role during efferocytosis as a scaffold protein modulating the activity of its associating proteins [72]. Since Ephexin4 was recognized as a new binding partner of Elmo1, the effects of Ephexin4 on efferocytosis has been elucidated. Ephexin4 indeed promotes the engulfment of apoptotic cells in a RhoG- and Rac1-dependent manner [35]. Interestingly, co-expression of Elmo1 and Ephexin4 synergistically promotes efferocytosis resulting from relief of the auto-inhibition of Ephexin4. The SH3 domain of Ephexin4 mediates homotypic intermolecular interaction, which generates an auto-inhibitory effect by the blockade of RhoG access to the DH domain of Ephexin4. This auto-inhibitory interaction is relieved by Elmo1; the N- and C-termini of Elmo1 bind to the DH and SH3 domain of Ephexin4, respectively. Therefore, the binding of Elmo1 to Ephexin4 relieves the auto-inhibition of Ephexin4 by disrupting the homotypic intermolecular interaction, leading to the augmented GEF activity of Ephexin4 toward RhoG and the consequent increase in the ability of efferocytosis [36,37]. It is notable that the regulatory mechanism for the activity of Ephexin4 differs from that of other Ephexins.

### 2.5. Ephexin5

Another name for Ephexin5 is Vsm-RhoGEF, standing for vascular smooth muscle-specific RhoGEF. As its name implies, it was identified as a RhoGEF expressed in vascular smooth muscle cells from different tissues including the heart, liver, kidney, aorta, and spleen. Along with the identification of its expression pattern, its roles as a key downstream regulator of EphA4 signaling have been revealed. Ephexin5 interacts with EphA4 through its DH-PH domain, and Ephrin-A-stimulated EphA4 phosphorylates Ephexin5, which activates RhoA and induces the formation of stress fibers related to the vascular smooth muscle contractility [18].

Ephexin5 is highly expressed in the brain, especially in the hippocampus, and its functions in the brain have been well established [22]. It is known that Eph-Ephrin signaling is important for the proper excitatory synapse formation in the brain. EphB binding to Ephrin-B enhances the kinase activity of EphBs in the developing dendrites, which results in the dendritic spine outgrowth and excitatory synapse formation. In the hippocampus, Ephexin5 shows a similar expression pattern to that of EphB2 and it interacts with EphB2 preferentially in neurons. Ephexin5 is a RhoA-specific GEF and decreases the number of excitatory synapses by activating RhoA. Upon Ephrin-B stimulation, however, activated EphB2 phosphorylates tyrosine 361 of Ephexin5 (mouse Ephexin5), leading to its ubiquitination by Ube3A and following proteasomal degradation. Therefore, the negative regulation of spine outgrowth by Ephexin5 is a checkpoint to limit the EphB-mediated excitatory synapse formation [22,73]. In addition, one recent study reported that Ephexin5 is required for spinogenesis. Ephexin5 is temporally increased at the site of new spine formation in hippocampal neurons prior to its removal and new spine formation. Interestingly, the complete loss of Ephexin5 prevents neuronal activity from promoting spinogenesis. Taken together, these data indicate that Ephexin5 may serve as a beacon locating sites of new spine formation keeping them in check until incoming activity promotes spine formation at these sites [74].

Angiogenesis occurs through a series of complicated processes including sprouting, migration, tubulogenesis, and stabilization [75]. To accomplish these processes, the actin cytoskeleton in endothelial cells is tightly controlled by various Rho GTPases. Vascular endothelial growth factor (VEGF) is one of the most crucial regulators for angiogenesis. The interaction between VEGF and its corresponding receptor, VEGFR2, facilitates the development of blood vessels via the activation of Cdc42 and the inactivation of RhoJ in the endothelial cells [76,77]. Ephexin5 has an important role in angiogenesis because it mediates the VEGF-induced Rho GTPase activity modulation. During angiogenesis, Ephexin5 is highly expressed in retinal endothelial cells where Ephexin5 promotes actin polymerization by regulating the activity of Cdc42 and RhoJ [34,78]. By increasing Cdc42 activity and decreasing RhoJ activity, Ephexin5 promotes actin polymerization in endothelial cells. The significance of Ephexin5 in angiogenesis is further supported by delayed retinal vascular growth in Ephexin5-deficient mice [34]. Furthermore, the mutations or aberrant expression of Ephexin5 are associated with cancer and neuronal disorders such as epilepsy, seizures and Alzheimer’s disease [38,79,80].

## 3. Targeting Ephexins in Diseases

All members of the Ephexin family function as GEFs for RhoA, Rac, Cdc42 and RhoG, which is closely related to cell proliferation and cell migration. Due to their pivotal roles in cell proliferation and migration, dysregulation of their activity or expression could be associated with tumorigenesis and metastasis. Indeed, the increased expression levels of Eph receptors or Ephexins have been observed in many cancer types. Moreover, mutations or transcript variants of Ephexins are also closely related to cancer [44,49,68]. Therefore, targeting the signal module or a regulator for the signal module could be a good therapeutic approach for cancer treatment.

A few studies tried to modulate the activity of Ephexin3 using peptides mimicking the inhibitory helix or the most highly conserved surface of the DH domain interacting the inhibitory helix. However, modulating their activity is only tested in in silico or in vitro systems [20,69,70]. Thus, the efficacy of the peptides should be further investigated in more physiologically relevant conditions. Another approach to target the signal module is to inhibit the activity of Src because the activity of Ephexin1 or Ephexin3 is modulated through phosphorylation by Src. A few studies showed that Ephexin1 expression is altered after spinal cord injury and treatment of PP2, a selective Src family inhibitor, for the injury leads to functional locomotor recovery [62,63]. Although PP2 is not specific to only Src, this approach shows a possibility that targeting an Ephexin regulator in the signal module could be effective therapeutics.

Furthermore, Ephexin1 and Ephexin5 are associated with neuronal disorders such as depression, epilepsy, and Alzheimer’s disease [38,61,79,80]. EphA4-Ephexin1 signaling was increased in mice after social defeat stress [61]. Through whole exome sequencing on children with epilepsy, a missense mutation (1810C to 1810T) in Ephexin5 was identified and this substitution resulted in ~50% reduction in its GEF activity on RhoA [79]. In addition, the amyloid-β accumulated brain showed an elevated Ephexin5 level and reduced postsynaptic spine density due to the decreased level of EphB2 [80]. Treatment of mice with depression-like phenotype with an EphA4 inhibitor showed an antidepressant-like effect [61]. These studies imply that it is also a good strategy to target the signal module for treatment of those neuronal disorders. Therefore, a drug modulating the Eph receptor-Ephexin signaling would be a promising treatment such disorders.

## 4. Conclusions

In this review, the biological significance of Ephexins in various physiological and pathological contexts and molecular mechanisms by which their GEF activity is regulated have been discussed. Starting from studies in the 2000s, their roles not only in neurogenesis but also in other cellular events have been reported. In particular, recent studies have provided the relevance of Ephexins to tumorigenesis and efferocytosis, suggesting that the unappreciated roles of Ephexins still remain and need to be explored. Accordingly, extensive and in-depth studies of them would provide a better understanding of the diverse processes, including neuronal development and cancer.

## Figures and Tables

**Figure 1 cells-08-00087-f001:**
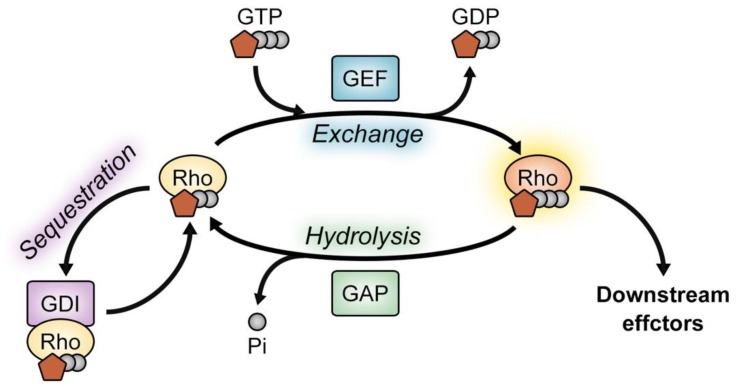
Overview of Rho GTPase regulation. The activity of Rho GTPases is controlled by GEFs, GAPs and GDIs. GEFs facilitate the exchange of GTPase-bound GDP for GTP but GAPs inactivate the Rho GTPase by hydrolyzing GTP. Additionally, the sequestration of Rho GTPases by GDIs modulates the level of active Rho GTPases. GEF, guanine nucleotide factor; GAP, GTPase-activating protein; GDI, guanine nucleotide dissociation inhibitor.

**Figure 2 cells-08-00087-f002:**
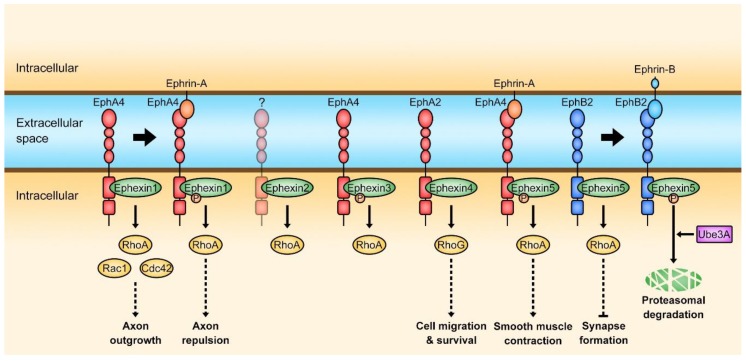
Ephrin-Eph receptor-Ephexin signaling. The activation of EphA4 by Ephrin-A increases the GEF activity of Ephexin1 toward RhoA whereas the EphB2 activation by Ephrin-B induces ubiquitination of Ephexin5 resulting in proteasomal degradation. An Eph receptor for Ephexin2 has not been reported.

**Figure 3 cells-08-00087-f003:**
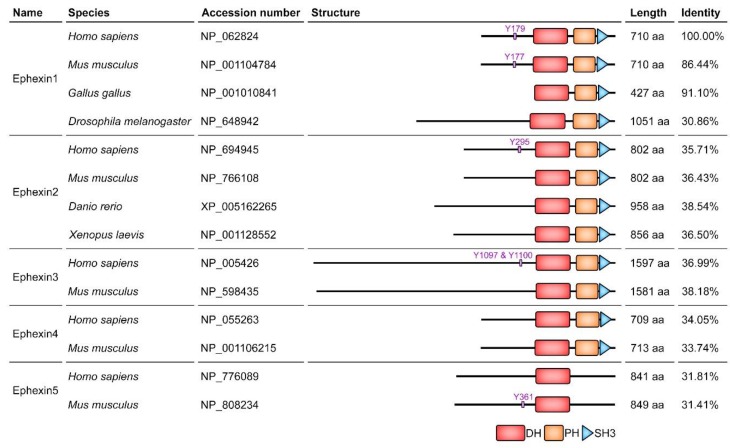
Schematic diagram of the members of the Ephexin family and sequence identity among homologs. The phosphorylation sites involved in alleviating the auto-inhibition are shown. The domains of Ephexins were structured according to SMART, a domain prediction program, and the sequence identity among homologs was calculated by Clustal Omega. DH, Dbl homology; PH, Pleckstrin homology; SH3, Src homology 3.

**Table 1 cells-08-00087-t001:** Overview of Ephexin family proteins.

Member	Aliases	Expression	GEF Specificity	Interacting Receptors	References
Ephexin1	Arhgef27, Ngef, Ephexin	Brain, spinal cord	RhoA, Rac1, Cdc42	EphA4	[19,54]
Ephexin2	Arhgef19, Wgef	Liver, kidney, heart, intestine	RhoA	– ^1^	[55]
Ephexin3	Arhgef5, Tim	Liver, kidney, colon, trachea, prostate, pancreas	RhoA, Rac1, Cdc42 ^2^	EphA4	[20,41,44]
Ephexin4	Arhgef16	– ^3^	RhoG, Cdc42	EphA2	[21,56,57]
Ephexin5	Arhgef15, Vsm-RhoGEF	Brain, vascular smooth muscles (liver, kidney, heart, spleen)	RhoA, Rac1, Cdc42 ^4^	EphA4, EphB2	[18,22,34]

^1^ There is no research that identifies the Ephexin2-interacting Eph receptors. ^2^ It is controversial whether Ephexin3 has GEF activity toward Rac1 and Cdc42 or not. ^3^ There is no research that directly deals with the expression pattern of Ephexin4. ^4^ It is controversial whether Ephexin5 has GEF activity toward Rac1 and Cdc42 or not.

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
