# Peer review of "Emerging Roles of Ephexins in Physiology and Disease"

_cells, 2019, doi:10.3390/cells8020087_

Round 1

Reviewer 1 Report

Emerging roles of Ephexins and their regulation.

Overall, this is a short and somewhat comprehensive quick overview of the Ephexin family of Rho GTPase exchange factors. Looking through the references there are only a few high profile studies on Ephexin molecules yet this review seems to lump all Ephexin studies together. I think this review could easily go a step further and highlighting some of the major discoveries on Ephexin molecules that are the strongest and most clear. I also think it would be important to create a table of reagents for studying the Ephexin molecules which would include a list of animals, cell lines, antibodies, expression constructs, and references. While I believe this review is making a good effort to highlight some aspects of Ephexins the authors miss an opportunity to provide some insight or perspective on what these molecules are doing in relation to development and disease. For example, it is not simply that Ephexin1 or Ephexin5 are highly expressed in the brain. Rather, these Ephexin molecules are expressed in the brain at particular times in development and possibly at distinct sub regions of the brain. Moreover, their misexpression can be pathological in various diseases. The location timing and tissue expression of the different Ephexins is not thoroughly discussed yet very relevant to understanding how they may regulate human physiology. Another important point when speaking about a family of molecule that have been studied using in vitro purified systems, cell lines, whole animals etc is to distinguish between these systems.  Ephexin molecules are not super conserved even in fly or chicken. Some critical regulatory domains don’t exist.  It is possible that the divergence of the sequences help to make these molecules specialized. Maybe by learning something about the Ephexin GEF and GAP protein codes we can engineer synthetic versions of such molecules to get them to perform specific tasks in specific tissues.  Just a thought that doesn’t need discussion in this review.  More to the point, I would like to hear a bit more from the authors what they are thinking. Another point, these molecules are regulated by phosphorylation and there is little discussion on this regulation. Yes, the Ephs are critical mediators of this but this leads to phosphorylation. What is phosphorylation doing? The authors should discuss a little more and provide identified sites of phosphorylation in their figure 2. Lastly, what are the next steps for Ephexin studies. This should be mentioned. Below I provide some other suggestions.

1)     At least one or two sentences are needed to introduce Eph receptor tyrosine kinases. There are also some good reviews that could be referenced.

2)     Line 47 to 48 there is a sentence that seems to be incomplete or unclear.  Please correct.

3)     Please define efferocytosis in the text.

4)     I do not like the word “sophisticatedly”.  Please consider another way to say this, as it is a bit confusing.

5)     The “Introduction” and “Rho GTPases and Rho Gefs” sections are a bit redundant. Please consider combining these sections.  Then go into the Ephexin section.

6)     Include accession numbers in the Ephexin figure so the readers know which Ephexin molecules the authors are speaking about (ie human or mouse etc.).

a.     If possible, include the predicted Ephexin proteins from other species and systems in figure 2. Include the accession numbers and refernces and % sequence identity to human Ephexin.

7)     Human Ephexin1 is about 700 aa while chicken is closer to 450 aa and Drosophila is about 1000 aa.  The authors should look at each of these sequences and determine whether the data for Ephexin1 in each of these systems is describing a role for Ephexin1 or Ephexin like molecules.  I think the way it is written a reader will interpret that Ephexin1 participates in each of the stated pathways regardless of system. This is not strictly correct and should be clarified. Just because one calls it Ephexin1 does not mean it is Ephexin1. Include a discussion about how these Ephexin molecules differ across species and then cautiously refer to drosophila or chicken Ephexin1 as dEphexin1 or cEphexin1.

8)     Wherever possible please include the system (cell line, in vitro purified, mouse, chicken, fly, etc.) where the described experiments were performed.  I do not think it is appropriate to lump all the data together to mean one thing. Also, some experiments are better than others. Referring to something as controversial is fine but without explanation is confusing. Why are the role for certain Ephexins in controlling specific GTPases controversial? What is the authors interpretation of the data? Where the experiments done the same way? Are they truly controversial or are there differences in experimental approach and quality of data that make interpretation difficult?  For example, different kits, antibodies, or tissue samples could have been used.  Please discuss.

9)     This sentence is confusing 263-265:

a.     The complete loss of Ephexin5 impedes the neural activity-dependent spinogenesis, and Ephexin5 is temporally increased at the site of new spine formation in hippocampal neurons [35]. I recommend: Emerging data indicate that Ephexin5 is temporally increased at the site of new spine formation in hippocampal neurons prior to its removal and new spine formation [35]. Interestingly, the loss of Ephexin5 prevents neuronal activity from promoting spinogenesis. Taken together, these data indicate that Ephexin5 may serve as a beacon locating sites of new spine formation keeping them in check until incoming activity promotes spine formation at these sites.  The absence of Ephexin5 leads to ectopic excess in spine formation and loss of activity dependent spine formation. This loss of activity induced spine formation is likely due to the fact that there are enough spines already present. Evidence to this point is that the Ephexin5 null mice appear to learn perfectly fine.

10)  There are only a few studies with the Ephexin knockouts look at animal behavior and human disease. These should be highlighted and discussed a bit more.

11)  Please include a brief section on targeting the Ephexin molecules to control disease. How would one do this? What are the considerations for specificity? One possibility is that the tissue and developomental timing of expression may make it possible to specifically target one Ephexin over another?

11) Provide information about whether Ephexin molecules are known to be mutated in human disease. If so, provide the sites of mutation for the specific Ephexin molecule.

Author Response

.

Reviewer 2 Report

Major suggestions

1.     My preference will go for a review discussing the emerging roles of Ephexins in physiology and disease. Therefore, the title can be modified to take into consideration the relevance of Ephexin in developmental processes, tissue homeostasis and pathological conditions where this family of molecules have been implicated. Emerging roles of Ephexins in Physiology and Disease could be a potential title for this review..

2.     The authors have done a great job highlighting what is known about each member of the Ephexin family. Following the above comment (1), I will suggest an in-depth discussions of the roles of Ephexin in normal physiological and developmental  processes, with a strong focus on what is known about member of Ephexin family in various pathological conditions (cancer, tumorigenesis, angiogenesis, immunity). See this paper for guidance on how to effectively organized this manuscript:  

Pasquale EB. Eph-ephrin bidirectional signaling in physiology and disease. Cell. 2008 Apr 4;133(1):38-52. doi: 10.1016/j.cell.2008.03.011. Review.

3.     In addition to Table 1 showing interaction between Ephexin and Eph receptors, a figure illustration of Eph/Ephexin interaction with downstream signaling molecules will strengthen this point.

4. Would it be possible to harness what is known on Ephexin signaling to help develop therapy for the treatment of pathological conditions associated with aberrant Ephexin signaling/expression? many be this point could be discussed at the conclusions.

Minor comments:

The writing style of this manuscript should be improved. May be proof-reading of the paper by a native English speaker is recommended.

Author Response

.

Round 2

Reviewer 2 Report

Minor comments mainly typos:

Line 139. “the eprhin-EphA4-Ephexin1 axis”. Correct: the ephrin-EphA4-Ephexin1 axis.

Line 176. “Ehexin2 are also limited” correct: Ephexin2 are also limited.

Line 208.”the correlation of Epheixn3/Src with cancer’ correct : the correlation of Ephexin3/Src with cancer

Line 209. promoting endothelial-mesenchymal transition (EMT). EMT is commonly used as an abbreviation for Epithelial-to-mesenchymal-transition and endothelial-to-mesenchymal-transition is abbreviated as EndoMT. Please correct.

Line 243. “the promoter region of Epheixn4 and…”  Correct: “ the promoter region of Ephexin4 and…”

Line 285.”one recent study reported that Ehpexin5…” Correct “one recent study reported that Ephexin5…”

Line 309-310. Indeed, an increase in the expression of Eph receptors and Ephexins constituting a signal module, Ephrin-Eph receptor-Ephexin, has been observed in many cancer types.  Please what is the meaning of this sentence? Do you want to say that Ephrin-Eph receptor-Ephexin form complex molecular cluster that is well conserved in various type of cancers?

Author Response

.
